# Six New Phenolic Glycosides from the Seeds of *Moringa oleifera* Lam. and Their *α*-Glucosidase Inhibitory Activity

**DOI:** 10.3390/molecules28176426

**Published:** 2023-09-04

**Authors:** Lin-Zhen Li, Liang Chen, Yang-Li Tu, Xiang-Jie Dai, Sheng-Jia Xiao, Jing-Shan Shi, Yong-Jun Li, Xiao-Sheng Yang

**Affiliations:** 1School of Pharmacy, Guizhou Medical University, Guiyang 550004, China; lilinzhen@gmc.edu.cn (L.-Z.L.); 13624538210@139.com (L.C.); tyltyl715@163.com (Y.-L.T.); daixiangjie2023@163.com (X.-J.D.); xshengjia213@163.com (S.-J.X.); 2Key Laboratory of Basic Pharmacology of Ministry of Education, Zunyi Medical University, Zunyi 563000, China; 3State Key Laboratory of Functions and Applications of Medicinal Plants, Guizhou Medical University, Guiyang 550004, China; yang_xiaosheng@yahoo.com; 4Engineering Research Center for the Development and Application of Ethnic Medicine and TCM, Ministry of Education, Guizhou Medical University, Guiyang 550004, China

**Keywords:** seeds of *Moringa oleifera* Lam, chemical constituents, phenolic glycosides, structure identification, *α*-glucosidase inhibition activity

## Abstract

Plant-derived phytochemicals have recently drawn interest in the prevention and treatment of diabetes mellitus (DM). The seeds of *Moringa oleifera* Lam. are widely used in food and herbal medicine for their health-promoting properties against various diseases, including DM, but many of their effective constituents are still unknown. In this study, 6 new phenolic glycosides, moringaside B–G (**1**–**6**), together with 10 known phenolic glycosides (**7**–**16**) were isolated from *M. oleifera* seeds. The structures were elucidated by 1D and 2D NMR spectroscopy and high-resolution electrospray ionization mass spectrometry (HR-ESI-MS) data analysis. The absolute configurations of compounds **2** and **3** were determined by electronic circular dichroism (ECD) calculations. Compounds **2** and **3** especially are combined with a 1,3-dioxocyclopentane moiety at the rhamnose group, which are rarely reported in phenolic glycoside backbones. A biosynthetic pathway of **2** and **3** was assumed. Moreover, all the isolated compounds were evaluated for their inhibitory activities against *α*-glucosidase. Compounds **4** and **16** exhibited marked activities with IC_50_ values of 382.8 ± 1.42 and 301.4 ± 6.22 μM, and the acarbose was the positive control with an IC_50_ value of 324.1 ± 4.99 μM. Compound **16** revealed better activity than acarbose.

## 1. Introduction

Diabetes mellitus (DM) is a metabolic disorder characterized by persistent hyperglycemia due to insufficient insulin secretion or impairment of islet function, which is now one of the major threats to human health in the 21st century [1]. Type II DM (T2DM, i.e., non-insulin-dependent DM) accounts for about 90% of the total DM patients in the current world [2]. Glycemic control is considered an effective therapy for the treatment of T2DM [3]. As is well-known, α-glucosidase is a carbohydrate hydrolase that acts on the terminal α (1→4) bonds of starch and disaccharides to release α-glucose in the brush border of the small intestine [4]. Through inhibiting the activities of α-glucosidase, the absorption of glucose in the intestine is slowed down, and the blood sugar level can be well managed [5]. Therefore, α-glucosidase inhibitors have become the focus of hypoglycemic drug research in recent years. At present, the most common α-glycosidase inhibitors are acarbose, voglibose, and miglitol. However, these inhibitors all have serious side effects, such as flatulence, abdominal cramping, and diarrhea [6]. Natural products are a rich source of safe and highly effective α-glucosidase inhibitors. Most of these natural bioactive compounds not only reduce hyperglycemia but are also associated with fewer side effects than currently applied α-glycosidase inhibitors and offer nutritional benefits for DM patients [7]. In recent years, a large number of studies have shown that compounds with α-glucosidase inhibitory activity have been screened from natural products [8].

*Moringa oleifera* Lam. belongs to the genus *Moringa* (family Moringaceae), native to the dry tropical forests of northwestern India [9]. *M. oleifera* is referred to as a “miracle tree” because of its rich nutritional and pharmacological properties [10]. It has high nutritional value, including protein, fiber, and a variety of vitamins, especially in seeds [11]. The seeds are rich in oils and unsaturated fatty acids, which can be used as a potential source of edible oil [12]. The seeds have many benefits for humans. These have aroused the interest of researchers. At present, there are few studies on the chemical constituents of the seeds; the biological activities are mainly directed to the crude extracts, and the pharmacodynamic material basis is not clear. To date, only several flavonoids, phenolic glycosides, and sterols [13,14,15] have been reported, which exhibit significant properties such as anti-hyperglycaemic, anti-inflammatory, anti-oxidation, and so on [16,17,18].

Herein, in the current study, we have studied the chemical constituents from an 85% ethanol extract of *M. oleifera* seeds and six new phenolic glycosides (**1**–**6**); ten known phenolic glycosides (**7**–**16**) (Figure 1) have been isolated and identified. All the secondary metabolites were evaluated for their inhibitory activities against *α*-glucosidase.

## 2. Results and Discussion

### 2.1. Structural Elucidation of the Isolated Compounds

The 85% EtOH extract from *M. oleifera* seeds was subjected to repeated column chromatography (CC) such as silica gel, Sephadex LH-20, Toyopearl HW-40F, and ODS, afforded six new phenolic glycosides (**1**–**6**) and ten known phenolic glycosides (**7**–**16**). Compounds **7**–**16** were identified as *O*-ethyl-4-[(4-*α*-L-rhamnosyl)-benzyl] carbamate (**7**) [19], 1-*O*-(4-hydroxymethylphenyl), *α*-L-rhamnopyranoside (**8**) [20], niazirin (**9**) [21], marumoside A (**10**) [22], niazimicin (**11**) [22], 4-aminophenol-*α*-L-rhamnopyranose (**12**) [23], moringa A (**13**) [24], 4-(*α*-L-rhamnosyloxy)benzylamine) (**14**) [25], *N*,*N′*-bis (4-[(*α*-L-rhamnosyloxy)benzyl]) thiourea (**15**) [26], and glucomoringin (**16**) [27] (Figure 1) by comparison of their spectroscopic data with those previously reported in the literature.

Compound **1** was obtained as a colorless viscous oil. Its molecular formula was determined as C_15_H_22_O_6_ by a positive high-resolution electrospray ionization mass spectrometry (HR-ESI-MS) ion at *m/z* 321.1302 [M + Na]^+^ (calculated for C_15_H_22_O_6_Na, 321.1309), indicating 5 degrees of unsaturation. The infrared (IR) spectrum showed absorption of hydroxyl (3392 cm^−1^), a methylene group (2915 cm^−1^), a benzene ring (1613, 1511 cm^−1^), and an ether bond (1231 cm^−1^). The ^1^H-nuclear magnetic resonance (NMR) spectrum of **1** revealed the presence of a 1,4-disubstituted benzene ring at *δ*_H_ 7.27 (2H, d, *J* = 8.5 Hz) and 7.04 (2H, d, *J* = 8.5 Hz), one ethoxy group hydrogen signal at *δ*_H_ 3.53 (2H, q) and 1.32–1.21 (3H, overlapped), one singlet methylene hydrogen signal at *δ*_H_ 4.43 (2H, s), and a rhamnose anomeric proton at *δ*_H_ 5.42 (1H, d, *J* = 1.6 Hz). The ^13^C NMR data showed 15 resonances, including 6 aromatic carbon signals at *δ*_C_ 157.4 (C-1), 133.4 (C-4), 130.4 (C-3, C-5), and 117.4 (C-2, C-6), 2 ethoxy carbon signals at *δ*_C_ 66.6 (C-8) and 15.4 (C-9), 1 methylene carbon signal at *δ*_C_ 73.2 (C-7), and 6 rhamnosyl carbon signals at *δ*_C_ 99.8 (C-1′), 73.8 (C-4′), 72.2 (C-3′), 72.0 (C-2′), 70.6 (C-5′), and 18.0 (C-6′). Furthermore, the location of the rhamnose at *δ*_C_ 157.4 (C-1) was verified by the ^1^H-detected heteronuclear multiple bond correlation (HMBC) spectrum correlation from *δ*_H_ 5.42 (1H, d, *J* = 1.6 Hz, H-1′) to C-1 (Figure 2). The chemical shifts and coupling constants of H-1′ indicated that the sugar is linked to the aglycone with *α*-glycosidic linkage. The singlet methylene signal at C-4 was confirmed by the HMBC correlation from *δ*_H_ 4.43 to C-4 and C-3. The HMBC correlations from *δ*_H_ 3.53 to C-7 indicated that the ethoxyl group was attached to C-7 (Figure 2). The gas chromatography (GC) analysis showed that the derivative of acid hydrolysis from **1** had the same retention time (*t*_R_ = 26.24 min) as the derivative of authentic L-rhamnose. Thus, compound **1** was identified as a phenolic glycoside derivative and named 4-(*α*-L-rhamnosyl) benzyl ethyl ester, which we trivially named moringaside B. All the ^1^H and ^13^C-NMR data of compound **1** were assigned (Table 1).

Compound **2** was obtained as a yellow oil. Its molecular formula was determined as C_21_H_30_O_10_ by positive HR-ESI-MS ion at *m/z* 443.1904 [M + H]^+^ (calculated for C_21_H_31_O_10_, 443.1912), indicating 7 degrees of unsaturation. The presence of a hydroxyl (3413 cm^−1^), a methylene group (2918 cm^−1^), a benzene ring (1613, 1514 cm^−1^), and an aromatic ether bond (1233 cm^−1^) was confirmed by its IR spectrum. The ^1^H-NMR spectrum showed an AA’BB’ coupling system aromatic ring at *δ*_H_ 7.39 (2H, d, *J* = 8.7 Hz) and 7.06 (2H, d, *J* = 8.7 Hz), an ethoxy group at *δ*_H_ 3.69 (1H, overlapped), 3.50 (1H, dq, *J* = 9.8, 7.1 Hz), and 1.19–1.22 (3H, overlapped), one methylene group at *δ*_H_ 5.83 (1H, s), and two sugar anomeric protons at *δ*_H_ 5.11 (1H, d, *J* = 1.5 Hz) and 5.45 (1H, d, *J* = 1.8 Hz). The ^13^C NMR spectrum showed that **2** had six aromatic carbon signals at *δ*_C_ 157.4 (C-1′), 133.4 (C-4′), 130.4 (C-3′, C-5′), and 117.4 (C-2′, C-6′), two ethoxy carbon signals at *δ*_C_ 63.7 (C-8) and 15.4 (C-9), one methylene carbon signal at *δ*_C_ 106.5 (C-7), and two groups of sugar carbon signals at *δ*_C_ 107.5 (C-1), 86.0 (C-2), 80.9 (C-3), 85.6 (C-4), 66.0 (C-5), 21.1 (C-6), and *δ*_C_ 99.7 (C-1″), 72.0 (C-2″), 72.2 (C-3″), 73.8 (C-4″), 70.7 (C-5″), and 18.0 (C-6″). Moreover, the HMBC correlation from *δ*_H_ 5.45 to C-1′ indicated that one sugar fragment group was attached to C-1′, and the *α*-configuration of anomeric carbon of the sugar fragment was deduced based on the coupling constant of the anomeric proton *δ*_H_ 5.45 (1H, d, *J* = 1.8 Hz). The location of the singlet methylene signal at C-4′ was confirmed by the HMBC correlation from *δ*_H_ 5.83 to C-4′, C-3′, and C-5′. Other sugars, C-2 and C-3, were attached to C-7 by oxygen atoms, respectively, which were confirmed by the HMBC correlation from *δ*_H_ 4.68 to C-7 and *δ*_H_ 4.89 to C-7 (Figure 2). The ^1^H-^1^H homonuclear chemical-shift correlated spectroscopy (COSY) spectrum correlations (Figure 2) of H-1/H-2, H-2/H-3, H-3/H-4, H-4/H-5, and H-5/H-6 showed the assignment in the protons of the sugar moiety. And this sugar was confirmed to *α*-configuration by the chemical shift and coupling constant of *δ*_H_ 5.11 (1H, d, *J* = 1.5 Hz). The GC analysis spectrum showed that the acid hydrolysate of **2** had the same retention time (*t*_R_ = 26.24 min) with the derivative of authentic sample L-rhamnose. The HMBC correlation from *δ*_H_ 3.70 (1H, m) and 3.50 (1H, m) to C-1, as well as the ^1^H-^1^H COSY correlation of H-8/H-9 was observed, suggesting that the ethoxyl group was located at C-1. The nuclear overhauser effect spectroscopy (NOESY) spectrum correlations of H-7 with H-1/H-5 and H-1 with H-3 revealed their co-facial relationship, and they were assigned arbitrarily as *α*-oriented, while the correlations of H-4 with H-6/H-2 indicated that these protons were *β*-oriented (Figure 3). The absolute configuration of C-1/C-2/C-3/C-4/C-5/C-7 was assigned as 1*R*/2*R*/3*S*/4*R*/5*R*/7*R* by comparing the calculated ECD data (Figure 4) with the experimental data. Consequently, compound **2** was identified as (1*R*, 2*R*, 3*S*, 4*R*, 5*R*, 7*R*)-*O*-ethly-2,3-*di*-*O*-(1′-*O*-*α*-L-Rha-phenylmethylene)-*α*-L-rhamnopyranoside and trivially named moringaside C. All the ^1^H and ^13^C-NMR data of compound **2** were assigned (Table 1).

Compound **3** was obtained as a yellow oil. Its molecular formula was determined as C_20_H_18_O_10_ by a positive HR-ESI-MS ion at *m/z* 451.1567 [M + Na]^+^ (calculated for C_20_H_28_O_10_Na, 451.1575), indicating 7 degrees of unsaturation. Its IR spectrum exhibited the presence of a hydroxyl (3413 cm^−1^), a methylene group (2918 cm^−1^), a benzene ring (1613, 1514 cm^−1^), and aromatic ether bond (1233 cm^−1^) functional groups. Analysis of the 1D and 2D NMR data of **3** were similar to those of compound **2,** except that the ethoxyl group attached to C-1 was replaced by methoxyl. The GC analysis showed that the acid hydrolysate of **3** had the same retention time as the standard L-rhamnose derivative (*t*_R_ = 26.24 min). Meanwhile, the chemical shift and coupling constant of the two sugar anomeric protons *δ*_H_ 5.45 (*J* = 1.8 Hz) and 5.00 (*J* = 1.5 Hz) proves that the two sugar segment groups are *α*-configuration. The NOESY spectrum correlation from H-7 to H-1/H-5 and H-1 to H-3 indicated their cofacial orientation (*α*-oriented), while the correlations from H-4 to H-6/H-2 suggested that these protons were *β*-oriented (Figure 3). The absolute configuration of C-1/C-2/C-3/C-4/C-5/C-7 was assigned as 1*R*/2*R*/3*S*/4*R*/5*R*/7*R* by comparing ECD spectra (Figure 4). Consequently, compound **3** was determined to be (1*R*, 2*R*, 3*S*, 4*R*, 5*R*, 7*R*)-*O*-methyl-2,3-*di*-*O*-(1′-*O*-*α*-L-Rha-phenylmethylene)-*α*-L-rhamnopyranoside, trivially named moringaside D. All the ^1^H and ^13^C-NMR data of compound **3** were assigned (Table 1).

Compound **4** was obtained as a yellow oil. Its molecular formula C_21_H_23_NO_6_ (11 degrees of unsaturation) was deduced by the ion peak of HR-ESI-MS *m/z* 408.1413 [M + Na]^+^ (calculated for C_21_H_23_NO_6_Na, 408.1423). The IR spectrum showed characteristic absorption peaks at 3383 cm^−1^, 2933 cm^−1^, 2256 cm^−1^, 1610 cm^−1^, 1508 cm^−1^, 1114 cm^−1^, 1062 cm^−1^, and 1026 cm^−1^, which were in agreement with a hydroxyl, a methylene group, a cyanogen group, a benzene ring, and an aromatic ether bond. The ^1^H NMR spectrum revealed AA’BB’coupling on a benzene ring at *δ*_H_ 7.14 (2H, d, *J* = 8.4 Hz, H-2, H-6) and 6.95 (2H, d, *J* = 8.4 Hz, H-3, H-5). Signals at *δ*_H_ 7.00 (1H, dd, *J* = 8.4, 2.4 Hz, H-5′), 6.97 (1H, d, *J* = 1.8 Hz, H-3′), and 6.78 (1H, d, *J* = 7.8 Hz, H-6′) showed that an ABX coupling system in an aromatic ring. Two methylene hydrogen signals were assigned at *δ*_H_ 3.87 (2H, s, H-7) and 3.69 (2H, s, H-7′). One anomeric doublet proton resonance at *δ*_H_ 5.36 (1H, d, *J* = 1.8 Hz, H-1″) showed the presence of one sugar unit as a glycoside. The ^13^C NMR spectrum of **4** showed 21 carbon signals including 12 aromatic carbon signals [*δ*_C_ 156.1 (C-1), 156.0 (C-1′), 136.2 (C-4), 131.2 (C-3′), 130.9 (C-3, C-5), 130.2 (C-2′), 127.9 (C-5′), 122.6 (C-4′), 117.5 (C-2, C-6), and 116.5 (C-6′)], 6 sugar carbon signals (*δ*_C_ 100.0 (C-1″), 73.9 (C-4″), 72.2 (C-3″), 72.1 (C-2″), 70.5 (C-5″) and 18.0 (C-6″)), 2 methylene carbon signals (*δ*_C_ 35.7 (C-7) and 22.7 (C-7′)), and 1 CN moiety signal at *δ*_C_ 120.2 (C-8′) [28]. The HMBC correlation from *δ*_H_ 5.36 to C-1 indicated that one sugar segment was attached to C-1, and according to the coupling constant of the anomeric proton *δ*_H_ 5.36 (*J* = 1.8 Hz), the *α*-configuration of the anomeric carbon of a sugar fragment was deduced. Furthermore, the GC analysis showed that the retention time of acid hydrolysate **4** was the same as the standard L-rhamnose derivative (*t*_R_ = 26.32 min), which indicated that **4** had *α*-L-rhamnose moiety. The methylene (C-7) linked to two benzene rings was confirmed by the HMBC correlation (Figure 2) of H-7 to C-4, C-1′, C-3′, and C-6′, and the downfield resonance of *δ*_C_ 156.0 (C-1′) suggested substitution of a hydroxyl residue. The location of the acetonitrile signal at the ABX coupling aromatic spin systems (C-4′) was confirmed by the key HMBC correlations from H-7′ to C-3′, C-4′, C-5′, and C-8′. Therefore, the structure of **4** was confirmed as 4-(*α*-L-rhamnopyranosyl) benzyl-1′-hydroxy-4′-phenylacetonitrile and trivially named moringaside E. All the ^1^H and ^13^C-NMR data of compound **4** were assigned (Table 2).

Compound **5** was obtained as yellow oil. Its negative HR-ESI-MS showed an [M - H]^−^ ion at *m*/*z* 424.1600, which is in accordance with the molecular formula C_20_H_27_O_9_N (calculated for C_20_H_26_O_9_N, 424.1602), indicating 8 degrees of unsaturation. The IR spectrum of 5 shows frequencies at 3394, 2933, 2252, 1612, 1510, 1236, 1064, and 1022 cm^−1^ and was assigned to a hydroxyl, a methylene group, a cyanogen group, a benzene ring, and an aromatic ether bond. The 1D and 2D NMR data of compound **5** showed a high degree of similarity to the compound 4-[(*β*-D-glucopyranosyl)-(1→3)-(*α*-L-rhamnopyranosyl)]phenylacetonitrile [29]. The only difference was that the C-3′ of compound **5** was attached to rhamnose by oxygen atoms, not glucose. Two sugar segment groups part of **5** were identified and characterized by the anomeric proton doublet at *δ*_H_ 5.49 (*J* = 1.8 Hz) and *δ*_H_ 4.75 (*J* = 1.2 Hz); these data suggested that sugars’ moieties were *α*-configuration. Additionally, acid hydrolysis of **5** obtained L-rhamnose, which was identified by the GC analysis comparison with authentic samples, which proved that **5** had *α*-L-rhamnose moieties. Consequently, the structure of compound **5** was confirmed as 4-[(*α*-L-rhamnopyranosyl)-(1→3)-(*α*-L-rhamnopyranosyl)]phenylacetonitrile and trivially named moringaside F. All the ^1^H and ^13^C-NMR data of compound **5** were assigned (Table 2).

Compound **6** was isolated as a yellow oil, and its molecular formula was assigned as C_21_H_30_O_11_ based on positive HR-ESI-MS data of the protonated species [M + Na]^+^ at *m/z* 481.1680 (calculated for 481.1685), indicating 7 degrees of unsaturation. Its IR spectrum exhibited the presence of hydroxyl (3385 cm^−1^), methylene group (2933 cm^−1^), carbonyl (1732 cm^−1^), and fatty ether bond (1064 cm^−1^, 1022 cm^−1^), with a comparison of the NMR data of **6** with the known compound methyl 2-[4-(*α*-L-rhamnopyranosyl)phenyl]acetate [21]. The differences were the addition of a group of sugar carbon signals (99.0, 72.8, 74.8, 73.7, 73.9, and 18.0) at C-3′ in **6**. This sugar fragment is attached at the C-3′ position, confirmed by *δ*_H_ 4.76, and has a correlation signal with C-3′ in the HMBC spectrum (Figure 2). The GC analysis showed that the acid hydrolysate of **6** was L-rhamnose; meanwhile, these two sugar moieties’ were *α*-configuration corroborated by the anomeric protons at *δ*_H_ 5.46 (*J* = 1.8 Hz) and 4.76 (*J* = 1.2 Hz). Eventually, compound **6** was elucidated as methyl 2-[4-(*α*-L-rhamnopyranosyl)-(1→3)-(*α*-L-rhamnopyranosyl)phenyl]acetate, which we trivially named moringaside G. All the ^1^H and ^13^C-NMR data of **6** were assigned (Table 2).

Compounds **1**–**6** are without precedent in the natural products literature, especially compounds **2**–**3**, which possess a rare 1, 3-dioxocyclopentane at the rhamnose group. A putative biosynthetic pathway for their scaffold is proposed in Figure 5. The 4-hydroxybenzaldehyde rhamnoside, a secondary metabolite of the seeds from *M. oleifera* [24], reacts with ethyl (or methyl) *α*-L rhamnoside in acidic conditions by acetal reaction to yield ethyl (or methyl) 2,3-*O*-benzylidene-*α*-L-rhamnopyranoside, which is synthesized in a pathway similar to that of methyl 2,3-*O*-benzylidene-*α*-D-mannopyranoside [30].

### 2.2. α-Glucosidase Inhibitory Activity Evaluation

α-glucosidase is a key catalytic enzyme for carbohydrate digestion and glucose release. Inhibition of α-glucosidase can delay glucose uptake and reduce postprandial blood glucose levels, which may inhibit the progression of DM [31]. Thus, all the isolated phenolic glycosides were evaluated for their inhibitory activities of *α*-glucosidase. As shown in Table 3, compared to the positive drug acarbose with an IC_50_ value of 324.1 ± 4.99 μM, compound **16** revealed excellent inhibitory activity of *α*-glucosidase with an IC_50_ value of 301.4 ± 6.22 μM, while compound **4** showed moderate activity with an IC_50_ value of 382.8 ± 1.42 μM. Other compounds had low inhibitory activity against *α*-glucosidase and are not listed in Table 3.

## 3. Material and Methods

### 3.1. General Experimental Procedure

The 1D and 2D NMR were recorded on a BRUKER 600 NEO NMR spectrometer (Bruker Co., Ltd., Karlsruhe, Germany) and a JEOL ECS 400 NMR spectrometer (Jeol, Tokyo, Japan). HR-ESI-MS data were measured using Thermo Fisher Q Exactive-Plus mass spectroscopy (Thermo Fisher Scientific, Waltham, MA, USA). GC Analysis was carried out on a Shimadzu-2010 Plus gas chromatograph (Shimadzu Co., Ltd., Kyoto, Japan). A JASCO J-715 spectrometer (Jasco, Tokyo, Japan) was used to record the ECD. UV spectra were acquired on a UV-2700 spectrometer (Shimadzu Co., Ltd., Kyoto, Japan). IR spectra were obtained using an IR Tracer-100 spectrometer (Shimadzu Co., Ltd., Kyoto, Japan). AMR-100 enzyme-linked immunosorbent assay (Hangzhou Aosen Instrument Co., Ltd., Hangzhou, China). Silica gel (200–300 mesh and 300–400 mesh, Qingdao Haiyang chemical Co., Ltd., Qingdao, China), Toyopearl HW-40F (Tosoh corporation, Tokyo, Japan), and Sephadex LH-20 (GE Healthcare Bio-Sciences AB, Uppsala, Sweden). The TLC were silica gel GF_254_ plates (Qingdao Haiyang Chemical Co., Ltd., Qingdao, China).

### 3.2. Plant Material

The seeds of *M. oleifera* were collected from Yunnan Province, People’s Republic of China, and identified by Shao-Huan Liu, a senior experimentalist at Guizhou Medical University. The voucher specimens were stored in the Engineering Research Center for the Development and Application of Ethnic Medicine and TCM (Ministry of Education), Guizhou Medical University, Guiyang, China.

### 3.3. Extraction and Isolation

The seeds of *M. oleifera* (10.5 kg) were slightly smashed and extracted with 85% ethanol under reflux, then concentrated the extract (2.5 kg). The extract was separated through a D101 macroporous resin, following elution in proper order by water, ethanol-water (30%, 60%, 95%, *v*/*v*). Finally, four parts were obtained: water part (1.8 kg), 30% ethanol part (276.7 g), 60% ethanol part (69.2 g), and 95% ethanol part (249.4 g).

The 60% ethanol extract was separated into 10 fractions (Fr. 1–10) through a silica gel column chromatography eluting with CH_2_Cl_2_–MeOH (100:0 to 1:1). Fr. 5 was separated by a silica gel CC eluting with CH_2_Cl_2_–MeOH (30:1 to 12:1) and HW-40F CC (MeOH) to obtain compounds **1** (23 mg) and **7** (90 mg). Fr. 6 was purified over Sephadex LH-20 (MeOH) and Toyopearl HW-40F (MeOH) to give compounds **8** (10.5 mg), **9** (9.8 mg), **10** (23 mg), and **11** (68.9 mg). Fr. 7 was chromatographed over silica gel (CH_2_Cl_2_–MeOH, 8:1 to 2:1) to get nine subfractions (Fr. 7.1–7.9). Fr. 7.6 was subjected to a Sephadex LH-20 (MeOH), Toyopearl HW-40F (MeOH), and ODS column to yield compounds **12** (34.2 mg), **13** (26.1 mg), **14** (26.7 mg), and **15** (470 mg). Fr. 10 was subjected to a silica gel CC eluting with EtOAc–MeOH (5:1 to 1:1) and an ODS column (MeOH/H_2_O, 1:9 to 7:3) to obtain compound **16** (79.0 mg). The 30% ethanol extract was separated into 11 fractions (Fr. 1–11) through a silica gel CC eluting with CH_2_Cl_2_–MeOH (100:0 to 1:1). Fr. 6 was separated by a silica gel CC eluting with EtOAc–MeOH (20:1 to 10:1) to get four subfractions (Fr. 6.1–6.7), and Fr. 6.3 was separated by Sephadex LH-20 (MeOH–Water, 1:1) to obtain four subfractions (Fr. 6.3.1–6.3.4). Fr. 6.3.2 was further separated by an ODS column to gain compounds **2** (5 mg) and **3** (7 mg). Fr. 6.3.3 was also chromatographed on an ODS column to yield compound **4** (8 mg). Fr. 8 was subjected to silica gel CC (EtOAc–MeOH, 20:0 to 8:1) to give seven sub-fractions (Fr. 8.1–8.7). Fr. 8.4 was applied to Sephadex LH-20 (MeOH–Water, 1:1) and an ODS column to get compound **5** (40 mg). Compound **6** (5.9 mg) was isolated from Fr. 8.5 by HW-40F CC (MeOH).

Moringaside B (**1**): colorless viscous oil; ${\left[\alpha \right]}_{D}^{25}$-90.8 (*c* 0.10, MeOH); UV (MeOH) λ_max_ (log *ε*) 206 (3.07), 224 (2.17) nm; IR (KBr) *ν*_max_: 3392, 2915, 1613, 1511, and 1231 cm^−1^; ^1^H and ^13^C NMR data, see Table 1; HR-ESI-MS: *m/z* 321.1302 [M + Na]^+^ (calculated for C_15_H_22_O_6_Na, 321.1309).Moringaside C (**2**): yellow oil; ${\left[\alpha \right]}_{D}^{25}$-111.3 (*c* 0.1, MeOH); UV (MeOH) λ_max_ (log *ε*) 206 (3.06), 224 (2.20) nm; IR (KBr) *ν*_max_: 3413, 2918, 1613, 1514, and 1233 cm^−1^; ECD (*c* 0.10, MeOH) Δε 214 (+0.63), 237 (-18.53) nm; ^1^H and ^13^C NMR data, see Table 1; HR-ESI-MS: *m/z* 443.1904 [M + H]^+^ (calculated for C_21_H_31_O_10,_ 443.1912).Moringaside D (**3**): yellow oil; ${\left[\alpha \right]}_{D}^{25}$-109.5 (*c* 0.1, MeOH); UV (MeOH) λ_max_ (log *ε*) 206 (3.10), 224 (2.25) nm; IR (KBr) *ν*_max_: 3396, 2913, 1612, 1512, and 1232 cm^−1^; ECD (*c* 0.08, MeOH) Δε 204 (-3.65), 228 (-10.90) nm; ^1^H and ^13^C NMR data, see Table 1; HR-ESI-MS *m/z* 451.1567 [M + Na]^+^ (calculated for C_20_H_28_O_10_Na, 451.1575).Moringaside E (**4**): yellow oil; ${\left[\alpha \right]}_{D}^{25}$-103.5 (*c* 0.1, MeOH); UV (MeOH) λ_max_ (log *ε*) 208 (3.28), 224 (2.21), and 280 nm (2.03) nm; IR (KBr) *ν*_max_: 3383, 2933, 2256, 1610, 1508, 1114, 1062, and 1026 cm^−1^; ^1^H and ^13^C NMR data, see Table 2; HR-ESI-MS *m/z* 408.1413 [M + Na]^+^ (calculated for C_21_H_23_NO_6_Na, 408.1423).Moringaside F (**5**): yellow oil; ${\left[\alpha \right]}_{D}^{25}$-146.2 (*c* 0.1, MeOH); UV (MeOH) λ_max_ (log *ε*) 222 (2.64), 272 (2.26) nm; IR (KBr) *ν*_max_: 3394, 2933, 2253, 1612, 1510, 1236, 1064, and 1022 cm^−1^; ^1^H and ^13^C NMR data, see Table 2; HR-ESI-MS *m/z* 424.1600 [M - H]^−^ (calculated for C_20_H_26_O_9_N, 424.1602).Moringaside G (**6**): yellow oil; ${\left[\alpha \right]}_{D}^{25}$-51.1 (*c* 0.1, MeOH); UV (MeOH) λ_max_ (log *ε*) 208 (3.18), 222 (2.44), and 272 (2.16) nm; IR (KBr) *ν*_max_: 3385, 2933, 1732, 1512, 1230, 1064, and 1022 cm^−1^;^1^H and ^13^C NMR data, see Table 2; HR-ESI-MS *m/z* 481.1680 [M + Na]^+^ (calculated for C_21_H_30_O_11_Na 481.1685).

### 3.4. Acid Hydrosis and Sugar Identification

The sugar was identified according to the established method [32]. Compounds **1**–**6** (each 0.3 mg), respectively, were dissolved with 2 mol/mL Hcl solution (2 mL) at 95 °C for 3 h. After cooling to room temperature, the ethyl acetate was added to the reaction solution and extracted three times. The water-soluble layer was dried to obtain the sugar residual. The sugar residuals, D-rhamnose (0.5 mL) and L-rhamnose (0.5 mL), separately, were added pyridine (0.4 mL) and L-cysteine methyl ester hydrochloride (1.0 mg), then heated at 60 °C for 1 h. N-trimethylsilyllimidazole (0.15 mL) was added to the mixture and reacted at 60 °C for 1 h again. Next, the reaction solution was dried, then dissolved in water (1.0 mL) and extracted with n-hexane (0.5 mL) three times. The organic layer was directly analyzed by GC analysis. The peaks of the acid-hydrolyzed derivatives of compounds **1**–**6** coincide with the derivatives of the authentic sample L-rhamnose.

### 3.5. Electronic Circular Dichroism Calculation of Compounds ***2**–**3***

The theoretical calculations were carried out using Gaussian 09 [33]. At first, all conformers were optimized at PM6. Room-temperature equilibrium populations were calculated according to the Boltzmann distribution law, based on which dominative conformers of population over 1% were kept. The chosen conformers were further optimized at B3LYP/6-31G(d) in the gas phase. Vibrational frequency analysis confirmed the stable structures. ECD calculations [34] were conducted at the B3LYP/6-311G(d,p) level in methanol with the IEFPCM model using the time-dependent density functional theory (TD-DFT). Rotatory strengths for a total of 10 excited states were calculated. The ECD spectrum was simulated in SpecDis by overlapping Gaussian functions for each transition according to Equation (1).
(1)$$\Delta \varepsilon (E)=\frac{1}{2.297\times 10^{-39}}\times \frac{1}{\sqrt{2\pi \sigma }}\sum_{i}^{A}\Delta E_{i}R_{i}e^{-{\left(\frac{E-E_{i}}{2\sigma }\right)}^{2}}$$

### 3.6. Inhibitory Activities against α-Glucosidase

The *α*-glucosidase enzyme inhibition assay was performed according to the previously described method [35]. Compounds **1**–1**6** were screened for *α*-glucosidase inhibitory activity with acarbose as a positive control (10 μg/mL) and DMSO as a blank control. In sequence, 10 μL of the sample, 100 μL of phosphate buffer (pH = 6.8), and 50 μL of *α*-glucosidase (0.5 U/mL) were added to 96-well plates and incubated for 15 min in a 37 °C incubator. Then, a further 40 μL of substrate (p-nitrophenyl-*β*-D-glucopyranoside, 1.25 mmol/L) was added and incubated in a 37 °C incubator for 25 min. After the reaction, the absorbance was measured at 405 nm. The results were obtained from at least three independent experiments.

## 4. Conclusions

In summary, we have conducted the successful isolation of 16 compounds from *M. oleifera* seeds, including 6 new and 10 known phenolic glycosides. Among them, compounds **2** and **3** especially are combined with a rare 1,3-dioxocyclopentane moiety at the rhamnose group. The putative biosynthetic pathway for their scaffold is proposed. All of the isolated compounds were evaluated for their inhibitory activities against *α*-glucosidase, and two compounds (**4** and **16**) showed comparable inhibitory activity to the positive control in vitro (Table 3). This study enriches the chemical basis of *M. oleifera* and elucidated the pharmacological basis of the hypoglycemic activity of the seeds. The results not only broaden the horizon of the structural diversity of phenolic glycosides of *M. oleifera* but also provide new evidence for the clinical applications of herbal medicine. Folk and ethnic medicines are of great importance and are valuable reservoirs for lead compounds in the field of drug research and development. From these results, further in-depth study may be done to discover the lead compounds.

## Figures and Tables

**Figure 1 molecules-28-06426-f001:**
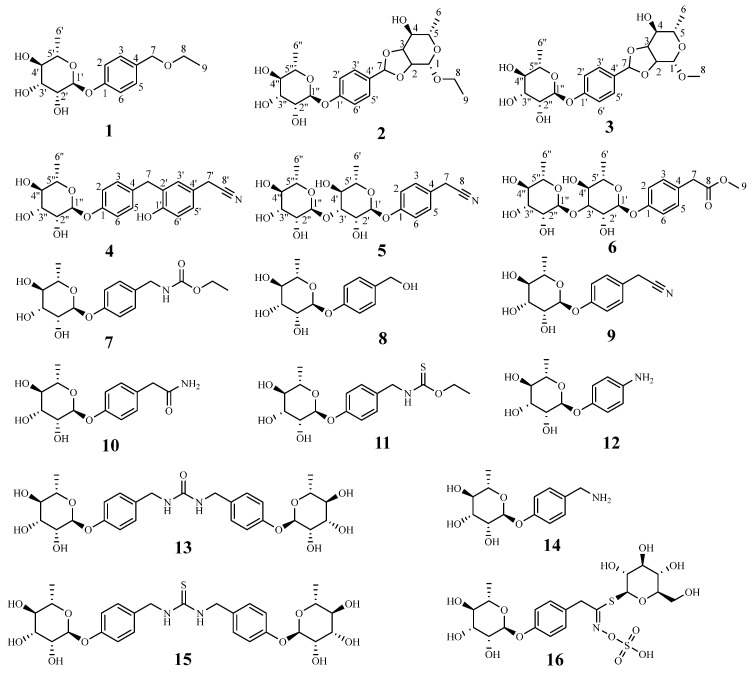
Chemical structures of compounds **1**–**16** isolated from *M. oleifera* seeds.

**Figure 2 molecules-28-06426-f002:**
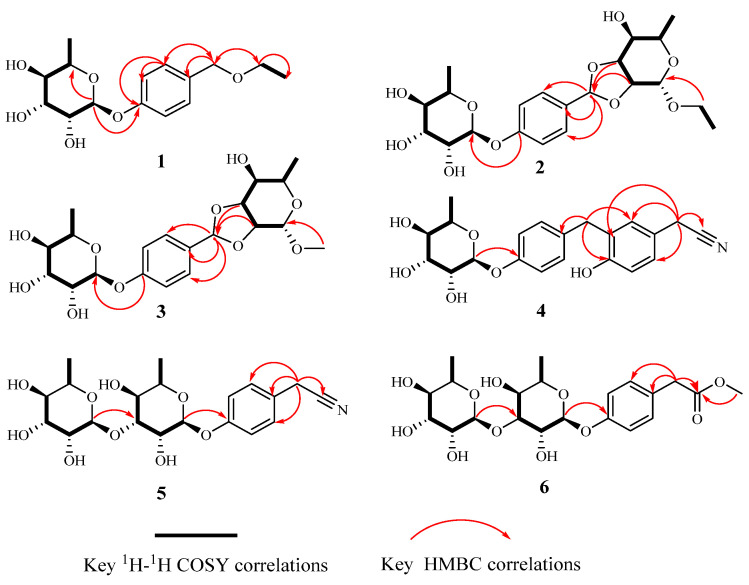
Key ^1^H-^1^H COSY and HMBC correlations of compounds **1**–**6**.

**Figure 3 molecules-28-06426-f003:**
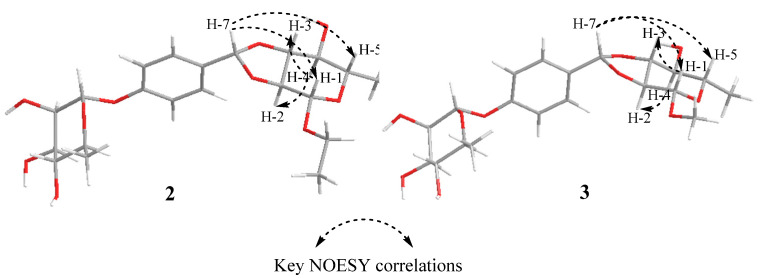
Key NOESY correlations for compounds **2**–**3**.

**Figure 4 molecules-28-06426-f004:**
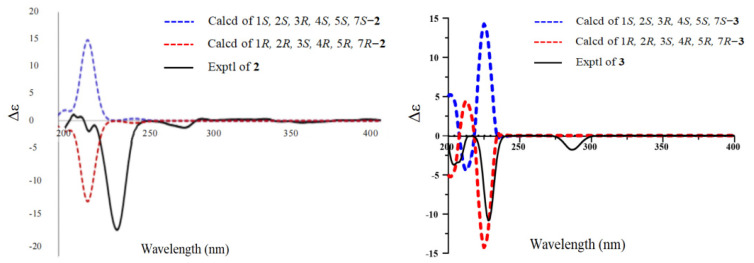
Calculated and experimental ECD spectra of compounds **2**–**3**.

**Figure 5 molecules-28-06426-f005:**
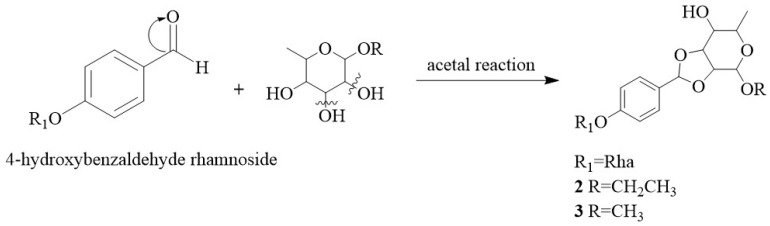
A putative biosynthetic pathway for compounds **2** and **3**.

**Table 1 molecules-28-06426-t001:** The ^1^H and ^13^C NMR spectroscopic data of compounds **1**–**3** in CD_3_OD.

NO.	1 ^a^		NO.	2 ^b^		3 ^b^	
	*δ*_H_ (*J* in Hz)	*δ* _C_		*δ*_H_ (*J* in Hz)	*δ* _C_	*δ*_H_ (*J* in Hz)	*δ* _C_
1	-	157.4	1	5.11 d (1.5)	107.5	5.00 d (1.5)	108.6
2, 6	7.27 d (8.5)	117.4	2	4.68 d (5.5)	86.0	4.68 d (5.4)	85.8
3, 5	7.04 d (8.5)	130.4	3	4.89 ov	80.9	4.89 ov	80.8
4	-	133.4	4	3.69 m	85.6	3.65 dd (9.0, 5.4)	85.6
7	4.43 s	73.2	5	4.07 dq (8.7, 6.3)	66.0	4.08 m	65.9
8	3.53 q (14.0, 7.0)	66.6	6	1.32 d (6.3)	21.1	1.33 d (6.0)	21.1
9	1.21–1.19 ov	15.4	7	5.83 s	106.5	5.83 s	106.5
1′	5.42 d (1.6)	99.8	8	3.70 m 3.50 dq (9.8, 7.1)	63.7	3.33 s	54.6
2′	4.00 dd (3.3, 1.9)	72.0	9	1.19–1.22 ov	15.4	-	-
3′	3.85 dd (9.5, 3.4)	72.2	1′	-	158.7	-	158.7
4′	3.46 t (9.5)	73.8	2′, 6′	7.06 d (8.7)	117.1	7.06 d (9.0)	117.1
5′	3.64 m	70.6	3′, 5′	7.39 d (8.7)	129.4	7.39 d (9.0)	129.4
6′	1.23–1.21 ov	18.0	4′	-	132.0	-	132.0
			1″	5.45 d (1.8)	99.7	5.45 d (1.8)	99.7
			2″	3.99 dd (3.5, 1.8)	72.0	3.99 dd (3.6, 1.8)	72.0
			3″	3.84 dd (9.5, 3.5)	72.2	3.84 dd (9.0, 3.0)	72.2
			4″	3.45 t (9.5)	73.8	3.45 t (9.6)	73.8
			5″	3.60 m	70.7	3.60 m	70.7
			6″	1.19–1.22 ov	18.0	1.21 d (6.0)	18.0

^a^: NMR data (*δ*) were measured at 400 MHz for ^1^H and 100 MHz for ^13^C; ^b^: NMR data (*δ*) were measured at 600 MHz for ^1^H and 150 MHz for ^13^C; ov: overlapping signals within the same column.

**Table 2 molecules-28-06426-t002:** The ^1^H (600 MHz) and ^13^C NMR (150 MHz) data of compounds **4**–**6** in CD_3_OD.

NO.	4		5		6	
	*δ*_H_ (*J* in Hz)	*δ* _C_	*δ*_H_ (*J* in Hz)	*δ* _C_	*δ*_H_ (*J* in Hz)	*δ* _C_
1	-	156.1	-	157.2	-	156.8
2, 6	6.95 d (8.4)	117.5	7.09 d (8.4)	118.1	7.04 d (8.4)	117.6
3, 5	7.14 d (8.4)	130.9	7.30 d (8.4)	130.4	7.22 d (8.4)	131.5
4	-	136.2	-	126.0	-	129.5
7	3.87 s	35.7	3.83 s	22.7	3.61 s	40.9
8	-	-	-	119.8	-	174.2
9	-	-	-	-	3.68 s	52.5
1′	-	156.0	5.49 d (1.8)	99.6	5.46 d (1.8)	99.7
2′	-	130.2	4.19 dd (1.8)	69.2	4.18 dd (3.2, 2.0)	63.8
3′	6.97 d (1.8)	131.2	4.04 dd (3.0)	79.7	4.05 dd (9.2, 3.6)	79.7
4′	-	122.6	3.60 t (9.0)	71.9	3.60 t (7.2)	72.0
5′	7.00 dd (8.4, 2.4)	127.9	3.67 m	70.3	3.70 m	70.2
6′	6.78 d (7.8)	116.5	1.24 d (6.0)	18.1	1.25 d (6.0)	18.1
7′	3.69 s	22.7	-	-	-	-
8′	-	120.2	-	-	-	-
1″	5.36 d (1.8)	100.0	4.75 d (1.2)	99.0	4.76 d (1.2)	99.0
2″	3.98 dd (3.6, 2.4)	72.1	4.00 dd (3.0)	72.7	4.00 d (3.6)	72.8
3″	3.83 dd (9.6, 3.6)	72.2	3.48 dd (3.0)	74.8	3.48 dd (6.0, 3.0)	74.8
4″	3.45 m	73.9	3.39 t (9.0)	73.6	3.40 t (9.0)	73.7
5″	3.66 t (9.6)	70.5	3.35 m	73.8	3.35 m	73.9
6″	1.22 d (6.6)	18.0	1.35 d (6.0)	18.0	1.36 d (6.0)	18.0

**Table 3 molecules-28-06426-t003:** *α*-glucosidase inhibitory activity of compounds **1**–**16** (*n* = 3).

Compound ^a^	IC_50_ (μM)
4	382.8 ± 1.42
16	301.4 ± 6.22
Acarbose ^b^	324.1 ± 4.99

^a^ Data of inactive compounds are not listed. ^b^ Positive control.

## Data Availability

All the data in this research were presented in the manuscript and Appendix A.

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
