# Peer review of "Six New Phenolic Glycosides from the Seeds of Moringa oleifera Lam. and Their α-Glucosidase Inhibitory Activity"

_molecules, 2023, doi:10.3390/molecules28176426_

Round 1

Reviewer 1 Report

The authors should include the full species name in the title: Moringa oleifera Lam.

The abstract provided by the authors is clear and well-structured. However, the biological part concerning glucosidase enzymes seems ambiguous, as if it were included merely to fill the manuscript. I recommend adding the importance of the isolated compounds in the abstract in diabetes treatment, along with their noteworthy conclusions. For instance, it should be emphasized that compound 4 exhibits better activity than acarbose. To support this, I highly recommend the authors read and cite the manuscript by Watanabe et al., 2021 (DOI: 10.3390/molecules26123513), a review on Moringa oleifera in diabetes.

The chemical structures in Figure 1 appear to be in low resolution. Please do not export each as an image to maintain the structure's resolution. Instead, use ChemDraw or similar software to create the template following ACS rules at 300 dpi for optimal resolution.

The manuscript's introduction is too brief and lacks a clear reflection of the work's objectives and research question that the authors intend to address. It appears overly focused on the chemical aspect, neglecting the biological part. Crucial questions, such as the authors' motivation for selecting glucosidases as a study model, are left unanswered. A quick search on PubMed revealed several review articles on this plant and its pharmacological applications, as well as secondary metabolites in its leaves. I suggest the authors conduct a more in-depth literature review and restructure their introduction accordingly.

The results and discussion section is well-organized, clearly describing the spectroscopic (NMR, IR, UV) and spectrometric (MS) processes, and well-supported by supplemental files. However, this section is entirely descriptive, needing a solid discussion based on the existing literature. Although the authors mentioned that the compounds are new, they do not elaborate on each compound's biosynthetic pathway, potential origins, critical enzymes involved, or related compounds in the literature that helped them solve the chemical structures. The biological part is also treated as an obligatory addition with minimal importance. The significance of evaluating glucosidase enzymes and their role in diabetes is not adequately discussed, nor are there comparisons with similar compounds. There is much room for improvement in the biological aspect. With significant effort, the authors can develop a more robust manuscript encompassing both areas.

The conclusions are too simplistic and lack any perspective. The authors have promising in vitro evaluations but need to highlight this information. This section needs to be restructured.

The manuscript presented by the authors is fascinating, as it introduces 16 novel molecules with activity against a-glucosidase inhibition. However, the authors have focused all their efforts on the chemical aspect while neglecting the biological part. Moreover, the manuscript is well-defined and descriptive, but a more exhaustive literature review is needed to improve the discussion. If the authors address these comments and make the necessary adjustments, the manuscript could be considered for publication in Molecules. However, at this stage, I think that the manuscript does not meet the criteria for publication and recommend that the authors make the indicated changes as mandatory major revisions.

Reviewer 2 Report

Dear authors, first of all, I would like to congratulate you on this valuable work. Researching the phenolic glycosides of the Moringa oleifera, which has started to be grown by sending culture species to many parts of the world, will provide a better understanding of the plant's potential. In this context, deciphering new phenolic glycosides in this plant indicates that you have created a unique academic output in the field. The methodology part of the study is compatible with the study content. The sources you have discussed are relevant to the subject and contain sufficient information. Your conclusions are consistent and meaningful with your findings.

However, you need to make some corrections. Please make the following corrections:

Line 42 - Place the details found at the end of the introduction, which is the material, method, or conclusion, in the methods and result section. In this section, explain the content of the study superficially, not in detail;

Line 58 - There is a font size error; please correct it;

Line 331 - write ''Among them'', remove ''of''.

Please make all the arrangements completely.

Regards.

Reviewer 3 Report

The presented work is a meticulous and appropriate description of six new phenolic glycosides, named moringaside B-G, along with 10 previously known phenolic glycosides in M. oleifera seeds. Interestingly, the chemical foundation of M. oleifera was further enhanced by this research, which also shed light on the pharmacological underpinnings of the seeds' hypoglycemic effects. In my opinion, the experiment's structure and the work's quality deserve to be accepted.

Round 2

Reviewer 1 Report

The authors have addressed each of my comments and provided a clear and precise response. Now, the title of the manuscript aligns with its content, and both the abstract and introduction have been significantly improved. The experimental section and the α-glucosidase model as an approach to combating diabetes are now clearly presented. The article's discussion is relevant and coherent with its results. The conclusions are appropriate. Based on these improvements, I recommend publishing the article in its current form.